# Wax Composition of Serbian *Dianthus* spp. (Caryophyllaceae): Identification of New Metabolites and Chemotaxonomic Implications [note 1]

**DOI:** 10.3390/plants12112094

**Published:** 2023-05-24

**Authors:** Marko Z. Mladenović, Milenko N. Ristić, Andrija I. Bogdanović, Novica R. Ristić, Fabio Boylan, Niko S. Radulović

**Affiliations:** 1Department of Chemistry, Faculty of Sciences and Mathematics, University of Niš, Višegradska 33, 18000 Niš, Serbia; markohem87@gmail.com; 2Faculty of Natural Science and Mathematics, University of Priština, Lole Ribara 29, 38220 Kosovska Mitrovica, Serbia; milenko.ristic@pr.ac.rs (M.N.R.); novica.ristic@pr.ac.rs (N.R.R.); 3Department of Biology and Ecology, Faculty of Sciences and Mathematics, University of Niš, Višegradska 33, 18000 Niš, Serbia; andrijab8@hotmail.com; 4School of Pharmacy and Pharmaceutical Sciences, Panoz Institute, and Trinity Biomedical Sciences Institute, Trinity College Dublin, D02 PN40 Dublin, Ireland

**Keywords:** *Dianthus*, wax, branched long-chain benzoates, β-diketones, triacontane-14,16-dione, dotriacontane-14,16-dione, tetratriacontane-16,18-dione, eicosyl tiglate, 30-methylhentriacontan-1-ol, chemotaxonomy

## Abstract

Although ethnopharmacologically renowned, wax constituents of *Dianthus* species were sporadically studied. A combination of GC-MS analysis, synthesis, and chemical transformations enabled the identification of 275 constituents of diethyl-ether washings of aerial parts and/or flowers of six *Dianthus* taxa (*Dianthus carthusianorum*, *D. deltoides*, *D. giganteus* subsp. *banaticus*, *D. integer* subsp. *minutiflorus*, *D. petraeus*, and *D. superbus*) and one *Petrorhagia* taxon (*P. prolifera*) from Serbia. Seventeen of these constituents (nonacosyl benzoate, additional 12 benzoates with *anteiso*-branched 1-alkanols, eicosyl tiglate, triacontane-14,16-dione, dotriacontane-14,16-dione, and tetratriacontane-16,18-dione) and two additional synthesized eicosyl esters (angelate and senecioate) represent completely new compounds. The structures of the tentatively identified β-ketones were confirmed by analysis of the mass fragmentation of the corresponding pyrazoles and silyl enol ethers obtained by transformations of crude extracts and extract fractions. Silylation allowed the identification of 114 additional constituents, including a completely new natural product (30-methylhentriacontan-1-ol). The results obtained by multivariate statistical analyses showed that the chemical profile of *Dianthus* taxa’s surface waxes is subject to both genetic and ecological factors, whereas the latter seemingly takes a more important role for the studied *Dianthus* samples.

## 1. Introduction

*Dianthus* L. is, after *Silene* L., the second largest genus in the Caryophyllaceae family with over 380 accepted species and 150 subspecies. The taxonomy of the genus has always been troublesome; it contains over 1000 synonyms and over 150 unresolved taxa. The position of genera *Petrorhagia* Link. and *Velezia* L. has been a subject of discussion by taxonomists for many decades, with many authors placing all species from *Velezia* within *Dianthus* and placing at least two *Petrorhagia* species in *Dianthus* [1]. *Dianthus* taxa are mostly perennial, rarely annual or biennial, herbs, and shrubs with decorative flowers, often fringed, speckled, and pleasantly perfumed, whose color ranges from white to shades of pink and purple [1]. Although a several species are ethnopharmacologically renowned, only a few taxa of this genus have been phytochemically studied in detail [2]. The chemical analyses revealed the presence of esters of benzoic and salicylic acids, triterpene saponins, and anthranilic acid-based phytoalexins, as well as other benzenoids, phenylpropanoids, isoprenoids, and nitrogen-containing compounds [2]. *Dianthus* taxa have a long history of use in the folk medicine of many nations; such use has been backed up by recent studies: a glycosylated flavonol (kaempferide triglycoside) isolated from *D. caryophyllus* and an extract of *D. chinensis* was found to be active against the human colon cancer cell line and hepatocellular carcinoma HepG2 cells, respectively [2]. Extracts of some *Dianthus* taxa were used in the treatment of gastrointestinal disorders, as well as anti-infective agents (gum infections, wounds, throat infections, etc.) [2].

The most widely used and accepted version of classification is that of Pax and Hoffmann, in which 7 sections are recognized and grouped into two subgenera: *D.* subg. *Armeriastrum* (Ser.) Pax & K.Hoffm. and *D.* subg. *Caryophyllum* (Ser.) Pax & K.Hoffm [3]. Both subgenera and all seven sections are represented by 36 species in the Serbian flora [4]. From a phytochemical point of view, the Serbian *Dianthus* taxa received little attention; more specifically the composition of waxes and volatiles of *D. cruentus* Griseb. was investigated previously resulting in the discovery of new plant metabolites [5].

In some instances, the chemical traits of plants proved to be more useful in the classification of taxa compared to the classical morpho-anatomical approach [6], opening a new niche in plant taxonomy—chemotaxonomy. Both approaches combined yield a much clearer insight into phylogenetic relationships, urging chemotaxonomical studies of taxa that proved problematic from only a morpho-anatomical standpoint, such as the infra- and intergeneric relationships of *Dianthus*. The chemical composition of epicuticular waxes was utilized for chemotaxonomical purposes, most frequently being limited to the alkane profiles [7,8,9]. Up to now, waxes of only a few species of *Dianthus* were analyzed including *D. caryophyllus* L. [10], *D. spiculifolius* Schur [11], and *D. cruentus* [5], finally, revealing even new long-chain compounds. The waxes of the closely related genus *Petrorhagia* have never been studied previously.

Prompted by the lack of phytochemical data on the composition of waxes of *Dianthus* species, in this work, we performed detailed chemical analyses of wax samples (diethyl-ether washings) of 6 *Dianthus* taxa (Table 1) and *Petrorhagia prolifera* (L.) P.W.Ball & Heywood (15 samples, in total) originating from Serbia. To achieve an unambiguous identification, besides an MS and RI comparison with the literature data, the identity of many wax constituents was confirmed by a GC co-injection with authentic samples (commercially obtained standards and synthesized compounds). The application of chemical transformations (dimethyl disulfide (DMDS) derivatization, silylation, and the synthesis of pyrazoles) performed directly on crude washings, chromatographic separations, and detailed mass spectral analysis of the derivatized wax samples allowed the identification of 389 wax constituents, among them 18 new natural products. To explore the possible applicability of the accumulated plant wax compositional data in the chemotaxonomy of *Dianthus* and related genera, we have performed multivariate statistical analyses (MVA) of wax compositional data for 7 different *Dianthus* taxa including the wax sample for one taxon that belongs to another closely related genus of the Caryophyllaceae family (*P. prolifera*), amounting, in total, to 16 statistically treated samples.

## 2. Results and Discussion

### 2.1. Composition of Washings and Their Variability

Detailed GC-MS analyses (a representative GC chromatogram of the washings is given in the case of *D. petraeus* in Appendix A), chemical transformation of the samples (DMDS derivatization and the synthesis of pyrazoles), and the synthesis of selected pure compounds enabled the identification of 275 constituents (Table 2) from fifteen samples of diethyl-ether washings of fresh aerial parts and/or flowers of 6 *Dianthus* taxa (*D. carthusianorum* (**1a**–**1d**), *D. deltoides* (**2a**–**2c**), *D. giganteus* subsp. *banaticus* (**3**), *D. integer* subsp. *minutiflorus* (**4**), *D. petraeus* (**5a**–**5c**), and *D. superbus* (**6a**–**6b**)) and one *Petrorhagia* taxon (*P. prolifera*, **7**). The identified constituents represented 87.2–99.0% of the total detected GC-peak areas (Table 2). The major compound classes were the long-chain alkanes (20.5–91.4%), β-diketones (up to 75.9%), aldehydes (0.3–38.7%), alkenes (0.9–28.2%), and esters (benzoates tr–16.7% and esters of normal/branched long-chain fatty acids or fatty alcohols (tr–19.1%)). The major constituents (Table 2) were heptacosane (up to 42.3%), nonacosane (3.5–26.4%), hentriacontane (3.2–44.4%), octacosanal (up to 20.1%), hexacosan-1-ol (up to 15.7%), hentriacontane-14,16-dione (up to 64.5%), and tritriacontane-16,18-dione (up to 18.6%). Alkenes with different double bond locations were identified by derivatization with dimethyl disulfide using the procedure that was previously described in detail [5].

Although most of the identified constituents represent ubiquitous wax constituents, that were also previously detected in the washings of other *Dianthus* taxa (e.g., *D. cruenthus* [5]), one homologous series of constituents caught our attention—esters of benzoic acid with long-chain (normal and branched) alcohols. Similar MS fragmentation patterns of 19 constituents (RI values 1374, 2213, 2315, 2418, 2521, 2625, 2728, 2832, 2936, 3040, 3143, 3247, 3350, 3453, 3556, 3660, 3765, 3870, and 3974) suggested that these constituents represent homologous benzoates of long-chain saturated *n*-alcohols. This reasoning was based on the characteristic MS fragmentation patterns (a base ion at *m*/*z* 123 (C_6_H_5_CO_2_H_2_^+^) and other intense ions at *m*/*z* 105 and 77 (C_7_H_5_O^+^ and C_6_H_5_^+^, respectively) which are indicative of benzoates (Figure 1). In some cases, the presence of the molecular ion allowed the allocation of the number of carbon atoms to the alcoholic moiety, but mostly it was the differences in the obtained RI values and a comparison of their retention indices with the literature values [5] that gave crucial information in their tentative identification. GC co-chromatography of the washings with a mixture of synthetic standards (obtained by esterification of benzoic acid with a mixture of *n*-alcohols) confirmed their presence in the washings. Additional benzoates (benzyl, 2-phenylethyl, (2*E*,6*E*)-2,6-farnesyl benzoate and docos-15-en-1-yl benzoate) were identified using the correlation of experimental RI data with available data from the literature, and in case of 2-phenylethyl and (2*E*,6*E*)-2,6-farnesyl benzoate by a co-injection experiment of the washings with synthesized pure esters.

Alongside the mentioned benzoates, a detailed analysis of the GC-MS partial current chromatograms (using *m*/*z* 123 and 105 ions) revealed the presence of two additional series (grouped based on their regular change of RI values; one with constituents eluting at RI = 2493, 2596, 2699, 3117, 3324, 3427, 3531, 3635, 3739, 3844, and 3948, and the second one with RI = 3210, 3417, 3625) of related benzoates that eluted slightly faster from the GC column compared to the *n*-chain homologs. Differences in the RI values with RIs of the normal chain isomers implied the existence of *iso-* and *anteiso-*branched-chains in the alcohol moiety [5]. Moreover, a comparison of the MS and RI data from the literature for 20-methylheneicosyl benzoate confirmed that one of the series represents an *iso*-branched series, whereas the constituents that eluted ca. 10 RI units slower represented the *anteiso*-branched counterparts [5]. Based on a detailed literature survey, 13 of the identified benzoates (Figure 2) represented new compounds, i.e., newly discovered natural products (Table 2).

Besides the abovementioned benzoates, a constituent at RI 2740 with the base ion at *m*/*z* 101 and intense peaks at *m*/*z* 83 and 55 was detected in the plant washings **2a**, **2b**, and **5b** (Table 2). This pointed to the presence of an ester of a branched pentenoic acid and a fatty alcohol. The presence of the molecular ion peak at *m*/*z* 380 suggested that the alcohol contained 20 carbon atoms. Esters of (*E*)-2-methyl-2-butenoic (tiglic), (*Z*)-2-methyl-2-butenoic (angelic), and 3-methyl-2-butenoic (senecioic) acids, and eicosan-1-ol were prepared. GC-MS co-chromatography of the synthetic standards with wax samples **2a**, **2b**, and **5b** agreed with the tentative RI/MS-based assumption, and the detected constituent was found to be eicosyl tiglate. According to a literature survey, all synthesized esters are completely new compounds whereas eicosyl tiglate represents a newly discovered natural product (Appendix A). The synthesized eicosyl esters were chromatographically (RI) and spectrally characterized (1D and 2D NMR and/or MS).

Interestingly, an additional group of identified constituents, *n*-chain fatty acid amides, and *N*-(2-phenylethyl) amides, were detected only in the sample of *D. giganteus* subsp. *banaticus* (**3**), could be regarded as compounds with a very restricted natural occurrence [12,13]. For example, both identified amides (*N*-(2-phenylethyl)eicosanamide and *N*-(2-phenylethyl)docosanamide; Table 2) were mentioned in the literature only once – as the metabolites of bacterial strains from the genus *Xenorhabdus* [13]. The mentioned amides are also excellent candidates for chemotaxonomic markers at species and even genus levels. However, detailed analysis of different populations of *D. giganteus* subsp. *banaticus* and other *Dianthus* species are needed to confirm this.

The second group of compounds that caught our attention was a series of GC-peaks, one of them being the major constituent of *D. petraeus* samples (Table 2), with a specific MS fragmentation pattern (base peak at *m*/*z* 100) indicative of long-chain β-diketones [14,15]. The differences in their retention index values (ΔRI ca. 100 units) and molecular ions suggested that they differ one from another in one -CH_2_- group (Table 2). Frequently, β-diketones, occurring in the leaf waxes of different plant species, were identified based on only fragmentation patterns visible in their mass spectra [14]. However, the position of the keto functionalities along the chain is difficult to be inferred from only a combination of MS and RI data due to their high similarity or inconclusiveness. Thus, some of the previous reports of β-diketones in plant waxes, with specific locations of keto groups, based only on GC–MS identification, should be taken with reserve. For instance, nonacosane-12,14-dione, triacontane-12,14-dione, and hentriacontane-14,16-dione were recently reported as constituents of plant cuticular waxes for four *Triticum aestivum* cultivars [16], but the authors provided no real proof of the exact regioisomeric nature of the detected compounds. The presented RI Thenacosane-12,14-dione (2689), triacontane-12,14-dione (2698), and hentriacontane-14,16-dione (2735) suggested that the proposed structures of the diketones should be revised [17].

### 2.2. The Analysis of GC Chromatograms after Chemical Transformations Performed Directly on Crude Wax Samples

Initially, to permit a definitive identification of the regioisomeric β-diketones, we turned to the derivatization reaction with hydrazine (Appendix A) [17]. A combination of MS data of the detected β-diketones (Appendix A) and the typical fragmentation pattern that was noted in the MS spectra of the synthesized corresponding pyrazole derivatives (e.g., pairs of peaks at *m*/*z* 277/264 and 305/292 ([C_18_H_33_N_2_]^+^/[C_17_H_32_N_2_]^+^ and [C_20_H_37_N_2_]^+^/[C_19_H_36_N_2_]^+^, respectively) for the pyrazole obtained from hentriacontane-14,16-dione; Appendix A) allowed us to unambiguously confirm the structure of the detected β-diketones as nonacosane-12,14-dione, hentriacontane-14,16-dione, and tritriacontane-16,18-dione [18]. A literature search showed that the above-mentioned β-diketones were only sporadically reported as plant/animal species metabolites and only one report included a *Dianthus* taxon or the plant family Caryophyllaceae in general [10]. The proposed structures of β-diketones were additionally confirmed by the silylation of the diethyl-ether washings (Appendix A). For example, the detected silylated enol forms of hentriacontane-14,16-dione, i.e., 14-((trimethylsilyl)oxy)hentriacont-14-en-16-one and 16-((trimethylsilyl)oxy)hentriacont-15-en-14-one, displayed the characteristic mass fragmentation pattern with intense peaks at *m*/*z* 325 [C_19_H_37_O_2_Si]^+^ and 353 [C_21_H_41_O_2_Si]^+^ (Appendix A) that, again, undoubtedly confirmed the position of β-diketone moiety and our tentative identification.

Unfortunately, due to the low abundance of the detected other homologous β-diketones at RI = 3297, 3498, 3698, and 3797 (Appendix A), it was only possible to predict the total number of C atoms in the molecules (30, 32, 34, and 35, respectively). However, the position of the β-diketone moiety remained unknown even after the derivatization of the crude extract samples as the silylated derivatives were not observed. For that reason, chromatographic separation of the wax sample was performed to obtain a fraction ‘rich’ in β-diketones. One of the seven different chromatographic fractions (fraction 3 from Appendix A that elute from the column with 7%, *v*/*v*, of the diethyl ether in hexane) displayed TIC peaks with more than 70% of the areas belonging to the homologous series of β-diketones. After the treatment of this fraction with hydrazine, partial ion current (PIC) chromatograms for *m*/*z* 96 [C_5_H_8_N_2_]^+^ and *m*/*z* 109 [C_6_H_9_N_2_]^+^ of the derivatized fraction revealed the presence of pyrazole derivatives of four additional β-diketones at Rt = 49.37, 52.03, 55.55, and 57.80 min with molecular ions peaks at *m*/*z* 446, 474, 502, and 516, respectively, besides the ones obtained from nonacosane-12,14-dione, hentriacontane-14,16-dione, and tritriacontane-16,18-dione (Rt = 48.19, 50.56, and 53.63; Appendix A). Further inspection of the mass spectra of these compounds revealed the presence of fragment ions that pointed to the position of the β-diketone moiety. For example, PIC chromatogram of *m*/*z* 292 and 305, corresponding to fragments [C_19_H_36_N_2_]^+^ and [C_20_H_37_N_2_]^+^, respectively, pointed that, besides nonacosane-12,14-dione, hentriacontane-14,16-dione, and tritriacontane-16,18-dione, two additional pyrazoles contained this pair of ions (Appendix A). These might be regarded as diagnostic fragment ions for [C_18_H_33_O_2_]^+^ moiety in the corresponding β-diketones, i.e., the position of the β-diketone moiety. Based on this and the presence of molecular ions peaks at *m*/*z* 502 and 516, we have assumed that the mentioned pyrazoles formed from tetratriacontane-16,18-dione, and pentatriacontane-16,18-dione. The presence of ion pairs (*m*/*z* 306/319 and 320/333 for pyrazole derivatives at Rt = 55.55, and 57.80 min, respectively) in the mass spectra additionally confirmed our identification. Additional PIC chromatograms of *m*/*z* 264 and 277, corresponding to the presence of fragments [C_17_H_31_N_2_]^+^ and [C_18_H_33_N_2_]^+^, respectively, revealed that pyrazoles at 49.37 and 52.03 min were derivatives obtained from triacontane-14,16-dione and dotriacontane-14,16-dione, respectively (Appendix A). The identified triacontane-14,16-dione, dotriacontane-14,16-dione, and tetratriacontane-16,18-dione represents new natural products and new compounds in general (Figure 3).

Unfortunately, the silylated fraction 3 did not contain unresolved peaks of the corresponding silyl enol ethers obtained from triacontane-14,16-dione, dotriacontane-14,16-dione, tetratriacontane-16,18-dione, and pentatriacontane-16,18-dione and for that reason, it was not possible to analyze their MS fragmentation pattern with certainty. Besides the structure confirmation of the new β-diketones, the chromatographic separation of *D. superbus* flower wax extract allowed the detection and identification of 97 constituents that were not detected in the GC-MS analysis of crude extract (denoted with minus in Appendix A).

Experimentally obtained RI data for β-diketones had an average increment of 100 units per CH_2_ in the series, which was in general agreement with those assigned to an *n*-alkane series. This might be explained by the low impact of the β-diketone moiety located somewhere in the middle of the molecule on RI values in such long-chain diketones, i.e., that a chain with more than twenty-nine carbon atoms is sufficient to make the RI increment of regioisomeric β-diketones essentially the same as for the *n*-alkane series. We believe that the proposed synthetic approach (pyrazole and silylenol ether formation) will make future identification of related natural compounds a straightforward task.

Besides the confirmation of proposed structures of β-diketones, analysis of the characteristic mass fragmentation of trimethylsilyl derivatives [19] and regularities in RI values enabled the identification of additional 114 wax constituents (Appendix A) that were not detected in the GC chromatograms of the original crude wax samples. Quite expectedly, additionally identified constituents belong to the (branched) long-chain carboxylic acids and alcohols. One group of compounds represents silyl ethers of homologous series of *n*-chain 1-alkanols (Appendix A). The identification was based on mass spectra and retention indices matching with the literature data. Detailed analysis of the silylated sample **5a** revealed the existence of an additional peak (Appendix A) that eluted slightly faster (Rt = 49.15 min) compared to 1-(trimethylsilyloxy)dotriacontane at Rt = 49.56 min. However, the mass fragmentation pattern (base ion at *m*/*z* 523), and the molecular ion peak (*m*/*z* 538) are almost identical to the already identified silyl derivative of 1-dotriacontanol (Appendix A). Thus, all these suggested that the mentioned compound could be a branched-chain alcohol with 32 carbon atoms. We assumed that the type of branching should be *iso*- or *anteiso*-alkan-1-ol due to biosynthetic considerations. The exclusion of other isomers (the presence of multiple branching, secondary alcohols, etc.) was inferred from the not-so-large differences in the RI values compared with the straight-chain isomer and from the different fragmentation patterns expected to be visible in their mass spectra [20]. Moreover, according to the data from the literature, the gas chromatographic behavior of *n*-, *iso*-, and *anteiso*- analog compounds, e.g., alkanols, esters, alkanes, etc., was more or less similar, i.e., *iso*- and *anteiso*-branched compounds have ca. 35- and 25-units lower RI values compared to the normal chain counterparts [5]. The difference between the RI of 1-docosanol and the detected alkanol was 37 units and that confirmed the presence of the *iso*-branched isomer of 1-docosanol, i.e., 30-methylhentriacontan-1-ol, which represents a completely new compound.

### 2.3. Multivariate Statistical Analysis (MVA) of Data Acquired from Untargeted GC-MS Metabolomics

To address the chemotaxonomical potential of the identified wax constituents, we decided to perform a statistical analysis of the up to now investigated *Dianthus* taxa, as well as one *Petrorhagia* species (Figure 4 and Figure 5 and Appendix A). We performed principal component analysis (PCA) and agglomerative hierarchical clustering (AHC). Both methods were applied utilizing two different variable sets: the original variables (constituent percentages that exceed 1% of total oil contribution in at least one of the samples) and sums of constituent classes (alkanes, alkenes, fatty acids, aldehydes, alcohols, benzoates, diterpenes, β-diketones, esters, other fatty acid related constituents, ketones, monoterpenes, sesquiterpenes, shikimate pathway metabolites, triterpenes, and unclassified constituents).

Based on the performed AHC using percentages of individual washings constituents as variables two well-separated clades could be recognized. Samples from the subgenus *Armeriastrum*, section *Carthusiani*, appear to show the highest degree of phenotypic plasticity when the wax composition is considered as they were scattered across the dendrogram (Figure 4), while other taxa formed more uniform groups. One of the clades consisted of solely *D. petraeus* samples and one *D. carthusianorum* sample. Interestingly, although these two taxa belong to distinct subgenera, this *D. carthusianorum* sample originated from the same collection locality as two samples of *D. petraeus* (Mt. Stara Planina), suggesting a profound effect of ecological conditions on the production of specific wax constituents (long-chain diketones). On the other hand, *D. deltoides* samples, similarly to *D. petraeus*, formed a closed group, although belonging to different sections but of the same subgenus (*Caryophyllum*). Once again it seems important to note that there was a higher level of similarity between samples coming from the same locality, as in the case of *D. superbus* and *D. deltoides* from Lake Vlasina, as opposed to the expected phylogenetic sectional grouping. Taxa belonging to the same section (*D. integer* and *D. petraeus*) were not placed within the same clades again pointing to the importance of ecological factors as predominant in the biosynthesis of wax constituents. The close relationship between *Petrorhagia* and *Dianthus* genera is further justified by the placement of *P. prolifera* sample amongst all other *Dianthus* samples. The composition of waxes from different plant organs (flowers and the rest of the aerial parts) seems to be more distinct one from another, in the cases of *D. superbus* and *D. carthusianorum* samples, implying a different biological function of the different organ waxes, and confirming the validity of the usual approach of comparing chemical compositions of same plant parts.

A comparable dendrogram (Appendix A) resulted when wax compound classes were used as variables, albeit with a higher degree of dissimilarity, deserving no further discussion. PCAs conducted with both sets of variables (Figure 5 and Appendix A) revealed a much more aggregated relationship between the samples, with only the flowers of *D. cruentus* and *D. superbus* as clearly distinct from the rest in the case of the biplot obtained with percentages of all washings’ constituents. The wax profile of *D. cruentus* was characterized uniquely by the presence of hexyl and other related alkyl esters of long-chain fatty acids (absent in all other analyzed *Dianthus* spp.). Although such esters are found in taxa outside *Dianthus* [21,22,23], they could be regarded as chemotaxonomic markers of *D. cruentus* since being exclusively found in this particular taxon and no other *Dianthus* spp. It appears that the washings of the highly fragrant flowers of *D. superbus* contained a significant number of volatiles compared to wax components and that they differentiated this sample from the rest.

Most of the identified wax constituents represent fatty acids-related compounds such as alkanes, β-diketones, alkenes, and esters. For that reason, strong dependencies between wax constituents in the Pearson matrix (expressed as correlation coefficients, r ≥ 0.9), obtained by principal component analysis (PCA; using original variables), were quite expected. All correlation coefficients within the group of major wax constituents, i.e., heptacosane, nonacosane, and hentriacontane had very high values (the amounts of these alkanes could be interconnected through the regulation of one or more enzymes that convert fatty acids to alkanes by elongation or decarbonylation). Other pairs of biosynthetically related compounds with high r values were the identified benzoates (e.g., dodecyl and tridecyl benzoate r = 1.000, 13-methylpentadecyl benzoate and eicosyl or docosyl benzoate r = 0.939 and 0.850, respectively, etc.). Additional high correlations were observed in the cases of several identified benzoates and some long-chain methyl esters (e.g., the correlation of octacosyl benzoate with methyl pentadecanoate, methyl linoleate, and methyl tetracosanoate was higher than 0.99) and between β-diketones and 2-ketones (e.g., the correlation of tritriacontan-16,18-dione with 2-pentatriacontanone, 2-tritriacontanone, and 2-hentriacontanone was 0.880, 0.893, and 0.733, respectively). These correlations suggested that the biosynthesis of such compounds is not only closely related but may involve either the same enzyme system or at least a common intermediate. Surprisingly, the relative content of hentriacontane-14,16-dione did not correlate with the relative amount of 16-hentriacontanone. It could be that the initially introduced ketone at position 14 undergoes an easy introduction of the second keto functionality in either β-positions, but if the ketone is initially introduced into position 16 it is not likely to be the subject of further oxygenation leading to this diketone.

However, the low discrimination between the majority of the samples, as visible from the bi-plot (Figure 5) obtained from the PCA could be the result of environmental factors producing sufficient background noise to prevent the expected taxonomic classification. Therefore, one should be rather cautious in reaching any chemotaxonomic conclusions from such analyses. We tried to overcome this limitation by subjecting supervised data to all MVA, more specifically, the contents of the constituents with a relative amount ≥ 2%, 3%, 5%, 10%, 15%, 20%, and 25% in at least one of the compared samples, to achieve a better chemotaxonomic classification. The obtained results were either identical or very similar to the ones presented in Figure 5 (the corresponding biplots are not shown for that reason). It follows that either other classification variables need to be introduced or a significantly higher number of samples (e.g., taxa) needs to be treated to reach the desired statistical result. When comparing the dendrograms obtained from a molecular biology study [1] with ours, sample sizes do not allow a meaningful interpretation and this is planned to be expanded in future studies.

## 3. Materials and Methods

### 3.1. General Experimental Procedures

All solvents were purchased from Sigma-Aldrich (St. Louis, MO, USA). Chemicals for synthetic use, including tiglic acid, angelic acid, senecioic acid, 1-eicosanol and series of *n*-chain 1-alkanols, benzoic acid, 4-(dimethylamino)pyridine (DMAP), *N*,*N*′-dicyclohexylcarbodiimide (DCC), hydrazine hydrochloride, dimethyl disulfide (DMDS), pyridine, *N*-methyl-*N*-(trimethylsilyl)trifluoroacetamide, and trimethylsilyl chloride were purchased from Sigma-Aldrich or Carl Roth (Karlsruhe, Germany). Silica gel 60, particle size distribution 40–63 mm (Acros Organics, Geel, Belgium), was used for dry-flash chromatography, whereas precoated Al silica gel plates (Merck (Darmstadt, Germany), Kieselgel 60 F_254_, 0.2 mm) were used for analytical TLC analyses. The spots on TLC were visualized by UV light (254 nm) and by spraying with 50% (*v*/*v*) aq. H_2_SO_4_ or 10% (*w*/*v*) ethanolic solution of phosphomolybdic acid, followed by 10 min of heating at 110 °C.

The GC-MS analyses (three repetitions) of the washings, derivatized washings, and pure synthesized esters were carried out using a Hewlett-Packard 6890N gas chromatograph equipped with a fused silica capillary column DB-5MS (5% diphenylsiloxane and 95% dimethylsiloxane, 30 m × 0.25 mm, film thickness 0.25 μm, Agilent Technologies, Palo Alto, CA, USA) and coupled with a 5975B mass selective detector from the same company. The injector and interface were operated at 250 °C and 320 °C, respectively. The oven temperature was raised from 70 °C to 315 °C at a heating rate of 5 °C/min and the program ended with an isothermal period of 30 min. As a carrier gas helium at 1.0 mL/min was used. The samples, 1.0 μL of the diethyl ether solutions (1.0 mg per 1.0 mL), were injected in a pulsed split mode (the flow was 1.5 mL/min for the first 0.5 min and then set to 1.0 mL/min throughout the remainder of the analysis; split ratio 40:1). MS conditions were as follows: ionization voltage of 70 eV, acquisition mass range 35–650, scan time 0.32 s.

The ^1^H- (including ^1^H-NMR spectra with homonuclear decoupling), ^13^C- (with and without heteronuclear decoupling) nuclear magnetic resonance (NMR) spectra, distortion less enhancement by polarization transfer (DEPT90 and DEPT135), and 2D (^1^H-^1^H COSY, NOESY, gHSQC, and gHMBC) NMR spectra of eicosyl tiglate were recorded on a Bruker Avance III 400 MHz NMR spectrometer (Fällanden, Switzerland; ^1^H at 400 MHz, ^13^C at 101 MHz) equipped with a 5–11 mm dual ^13^C/^1^H probe head. All NMR spectra were measured at 25 °C in CDCl_3_ with tetramethylsilane (TMS) as an internal standard. Chemical shifts are reported in ppm (*δ*) and referenced to TMS (*δ*_H_ = 0 ppm) in ^1^H-NMR spectra and/or to solvent (deuterated chloroform: *δ*_H_ = 7.26 ppm and *δ*_C_ = 77.16 ppm) in ^13^C- and heteronuclear 2D spectra. Scalar couplings are reported in hertz (Hz).

### 3.2. Plant Material

Fresh aerial parts and/or flowers of *Dianthus* taxa originated from the high slopes of Mt. Kopaonik, Mt. Šara, Mt. Suva Planina, and Mt. Stara Planina, as well as near Lake Vlasina and Deliblatska peščara (Table 3). Voucher specimens were deposited in the Herbarium of the Faculty of Sciences and Mathematics, University of Niš, Serbia. The identity of the plant material was determined by one of the authors (A.I.B) and the late professor of botany Vladimir Ranđelović (Department of Biology and Ecology, Faculty of Sciences and Mathematics, University of Niš).

### 3.3. Preparation of Plant Washings

Fresh aerial parts or flowers of mentioned *Dianthus* taxa, handled one by one, were shortly (*ca*. 5 s) immersed in a vessel with 500 mL of diethyl ether, while being exposed to ultrasonic waves (the glass beaker was inside an ultrasonic bath, Elmasonic S30 (Elma, Germany) operating at a frequency of 37 kHz, with an effective ultrasonic power of 30 W and a peak of 240 W), at room temperature. To remove all the insoluble material, the washings were gravity filtered through a small column packed with several grams of Celite^®^ (Merck, Germany), and dried over anhydrous MgSO_4_, then concentrated to 10 mL at room temperature before GC-MS analysis. The yield of the washings (%, *w*/*w*) was 0.11–0.37%.

### 3.4. Chromatographic Fractionation of Crude Flower D. superbus Washings *(**6a**)*

*D. superbus* flower washings were subjected to dry-flash column chromatography using a gradient of diethyl ether (Et_2_O) and *n*-hexane (from pure *n*-hexane to pure Et_2_O, with an increment step of 5%, *v*/*v*; fraction volume: 100 mL) and this resulted in 7 different fractions, in total, pooled based on TLC and GC-MS analyses (see Appendix A).

### 3.5. Component Identification

Diethyl-ether washings constituents were identified by comparison of their linear retention indices (relative to *n*-alkanes on a DB-5MS column) with the literature values, ΔRI values for the corresponding branched-chain isomers, their mass spectra with those of authentic standards, as well as those from Wiley 7, NIST14, MassFinder 2.3, and a homemade MS library with the spectra corresponding to pure substances, and, wherever possible, by co-injection with an authentic sample (see Table 2 and Appendix A; column ID). Additionally, samples of the washings and selected washings fractions were subjected to derivatization reactions that included reaction with hydrazine, silylation, and formation of dimethyl disulfide adducts, described in detail below, and afterward to additional GC-MS analyses.

### 3.6. Synthesis of Esters

A solution of 1-eicosanol (200 mg), (*E*)-2-methyl-2-butenoic acid (syn. tiglic acid; 1.1 eq), (*Z*)-2-methyl-2-butenoic acid (syn. angelic acid; 1.1 eq) or 3-methyl-3-butenoic acid (syn. senecioic acid; 1.1 eq), DMAP (0.3 eq) and DCC (1.1 eq) in 10 mL of dry CH_2_Cl_2_ (Appendix A) was stirred overnight at room temperature [5]. The crude esters were purified by dry-flash chromatography on silica gel using 3% (*v*/*v*) diethyl ether in hexane. The purity of the esters was checked by TLC and GC-MS. Mass spectra, 1D, and 2D NMR spectra of the new natural product, eicosyl tiglate, are given in the Appendix A. NMR spectral and/or GC-MS data for synthesized esters are given below:

Eicosyl tiglate: yield 88%; white waxy solid. RI (DB-5MS): 2740. MS (EI), (*m*/*z*, (relative abundance, %)): 380(M^+^, 1), 111(5), 102(10), 101(100), 100(25), 97(9), 85(6), 84(5), 83(30), 82(6), 71(7), 70(4), 69(10), 57(14), 56(5), 55(24), 43(15), 41(9). ^1^H-NMR (400 MHz, CDCl_3_): 6.84 (quartet of quartets, *J* = 7.0, 1.2 Hz, H-(C3), 1H), 4.12 (triplet, *J* = 6.7 Hz, H-(C1′), 2H), 1.83 (pseudo quintet, *J* = 1.2 Hz, H-(C5), 3H), 1.79 (doublet of quartets, *J* = 7.0, 1.2 Hz, H-(C4), 3H), 1.66 (quintet, *J* = 6.7 Hz, H-(C2′), 2H), 1.43—1.19 (overlapping signals, H-(C3’)—H-(C19′), 34H), 0.88 (pseudo triplet, *J* = 6.7, H-(C20′). ^13^C-NMR (101 MHz, CDCl_3_): 168.2 (C1), 136.8 (C3), 128.8 (C2), 64.6 (C1′), 31.9, 29.7, 29.7, 29.6, 29.6, 29.4, 29.3, 26.1, 22.7 (C(3′)—C(19′)), 28.7 (C2′), 14.3 (C5), 14.1 (C20′), 12.0 (C4).

Eicosyl angelate: yield 65%; white waxy solid. RI (DB-5MS): 2682. MS (EI), (*m*/*z*, (relative abundance, %)): 380(M^+^, 3), 111(3), 102(6), 101(49), 100(100), 97(6), 85(8), 84(3), 83(23), 82(6), 71(10), 70(4), 69(10), 57(18), 56(5), 55(25), 43(16), 41(10).

Eicosyl senecioate: yield 82%; white waxy solid. RI (DB-5MS): 2722. MS (EI), (*m*/*z*, (relative abundance, %)): 380(M^+^, 5), 111(3), 102(6), 101(85), 100(100), 97(7), 85(10), 84(5), 83(60), 82(10), 81(3), 71(10), 70(4), 69(11), 67(4), 57(20), 56(6), 55(27), 54(3), 43(23), 41(14).

### 3.7. Silylation Procedure

A sample of washings (*ca*. 10 mg) was placed into a GC vial, afterward, 500 μL of pyridine, 100 μL of *N*-methyl-*N*-(trimethylsilyl)trifluoroacetamide, and one drop of trimethylsilyl chloride were added (Appendix A). The vial was capped and heated for 1 h at 60 °C in a heating block [24]. After cooling to room temperature, 5 μL of the pyridine solution of TMS derivatives was injected; the GC-MS program used for this purpose was identical to the one used to record the original samples before silylation but also included a 7 min delay (the time that elapsed after the injection until MS detector turns on).

### 3.8. Dimethyl Disulfide (DMDS) Derivatization

The sample of the washings was dissolved in DMDS (0.25 mL per mg of the sample) and a solution (0.05 mL per mg of the sample) of iodine in diethyl ether (60 mg/mL) was added [5]. The mixture was stirred at room temperature overnight (Appendix A). Then diethyl ether was added, and the obtained mixture was washed with 10% aq. Na_2_S_2_O_3_, was dried over anhydrous MgSO_4_, and evaporated to dryness. The residue was taken up in fresh diethyl ether and directly analyzed by GC-MS.

### 3.9. Synthesis of Pyrazoles

A sample of the washings (*ca*. 10 mg) or the chromatographic fraction (see Appendix A, fraction F3 from Appendix A) was placed into a GC vial, afterwards, 500 μL of absolute ethanol and 100 μL of 98+ hydrazine monohydrate was added [17]. The vial was heated for 1 h at 60 °C in a heating block (Appendix A). After cooling to room temperature, the obtained sample was directly analyzed by GC-MS.

### 3.10. Multivariate Statistical Analysis

Principal component analysis (PCA) and agglomerative hierarchical clustering (AHC) were performed using the Excel program plug-in XLSTAT version 2022.5.1. The proximity between two objects in the AHC was measured by the Euclidean distance. Ward’s method was applied as the aggregation criterion. The number of object classes (groups of observations) was chosen based on the increase of within-group and between-group dissimilarities. PCAs of Pearson (n)-type were performed. Both methods were applied utilizing four different types of variables, i.e., original variables (contents of the constituents with a relative amount ≥ 1% in at least one of the compared samples) and transformed variables—summed-up contents of constituent classes (alkanes, alkenes, fatty acids, aldehydes, alcohols, benzoates, diterpenes, β-diketones, esters, other fatty acid related constituents, ketones, monoterpenes, sesquiterpenes, shikimate pathway metabolites, triterpenes, and unclassified constituents).

## 4. Conclusions

Comprehensive GC-MS analysis of fifteen samples of diethyl-ether washings obtained from fresh aerial parts and/or flowers of six *Dianthus* taxa and one *Petrorhagia* taxon from Serbia enabled the identification of 275 constituents. Among them, 18 identified constituents represent completely new compounds: nonacosyl benzoate, additional 12 benzoates with *anteiso*-branched long-chain 1-alkanols, eicosyl tiglate, 30-methylhentriacontan-1-ol, triacontane-14,16-dione, dotriacontane-14,16-dione, and tetratriacontane-16,18-dione. The used approach implies that the identification necessitates the synthesis of pyrazoles and silylenol ethers as complementary identification methods as RI data and fragmentation patterns in the mass spectra are practically indistinguishable for certain regioisomeric species within these series. The results of the MVA show that the chemical traits of surface waxes of *Dianthus* taxa are subject to both genetic and ecological factors, whereas the latter seemingly take a more important role for the studied *Dianthus* samples. As the taxa from the genus *Dianthus* have been demonstrated to show extremely low genetic diversity [1], this phenotypic plasticity of waxes, apparent for the MVA analyses, was rather unexpected.

## Figures and Tables

**Figure 1 plants-12-02094-f001:**
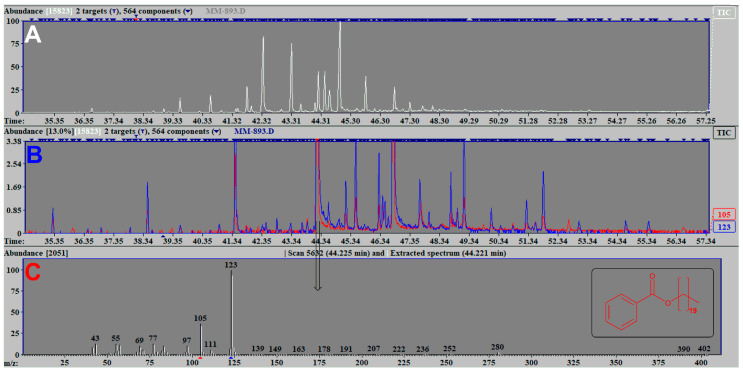
(**A**)—Total ion current chromatogram (TIC) of diethyl-ether washings of *D. deltoides* (sample **2a** from Table 2), (**B**)—corresponding patrial ion current chromatogram (PIC, ions *m*/*z* 105 and 123) and (**C**)—mass spectrum (MS) of eicosyl benzoate detected at (R_t_ 44.225 min).

**Figure 2 plants-12-02094-f002:**
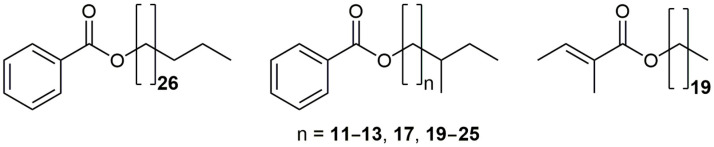
Structures of the newly identified wax esters, from left to right: nonacosyl benzoate, 12 benzoates of *anteiso*-branched 1-alkanols, and eicosyl tiglate.

**Figure 3 plants-12-02094-f003:**
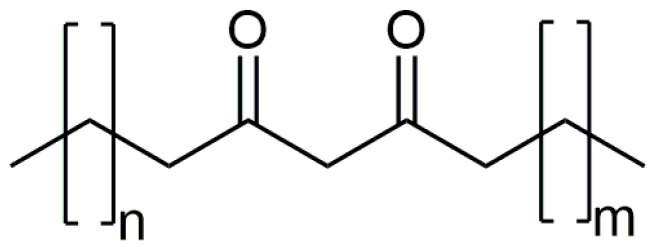
Structures of the newly identified β-diketones: triacontane-14,16-dione (n = 11 and m = 12), dotriacontane-14,16-dione (n = 11 and m = 14), and tetratriacontane-16,18-dione (n = 13 and m = 14).

**Figure 4 plants-12-02094-f004:**
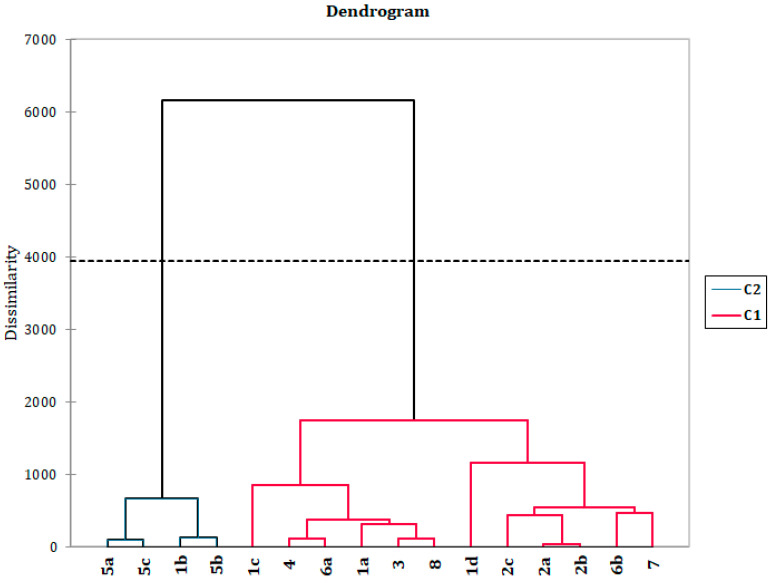
Dendrogram of AHC obtained by agglomerative hierarchical clustering using the original variables (contents of constituents with relative amounts ≥ 1% in at least one of the compared samples) and representing the chemical-composition dissimilarity relationships of 16 wax samples (observations) of 7 different *Dianthus* taxa (15 samples) and one sample of other Caryophyllaceae species (*Petrorhagia prolifera*). As a dissimilarity metric, the Euclidian distance was used (dissimilarity within the interval [0, 3900], using Ward‘s method as an aggregation criterion). Two statistically different groups of oils were found (**C1**–**C2**).

**Figure 5 plants-12-02094-f005:**
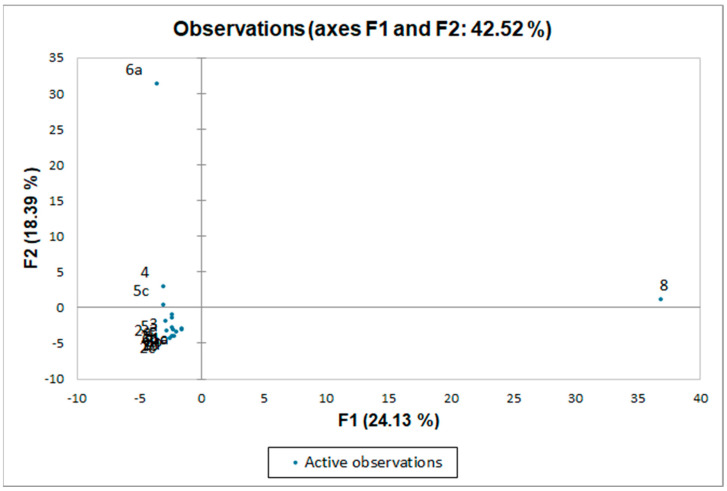
Dendrogram of PCA obtained by principal component analysis using the original variables (contents of constituents with relative amounts ≥ 1% in at least one of the compared samples) and representing the chemical-composition dissimilarity relationships of 16 wax samples (observations) of 7 different *Dianthus* taxa (15 samples) and one sample of *Petrorhagia prolifera*.

**Table 1 plants-12-02094-t001:** Analyzed plant taxa and their position within the genus.

*Dianthus* Taxa
*D.* subg. ***Armeriastrum*** (Ser.) Pax & K.Hoffm.
*D.* sect. *Carthusiani* (Boiss.) F.N.Williams
*Dianthus carthusianorum* L.
*Dianthus giganteus* subsp. *banaticus* (Heuff.) Tutin
*D.* subg. ***Caryophyllum*** (Ser.) Pax & K.Hoffm.
*D.* sect. *Barbulatum* F.N.Williams
*Dianthus deltoides* L.
*D.* sect. “*Tetralepides Leiopetala*” F.N.Williams
*Dianthus integer* subsp. *minutiflorus* (Halácsy) Bornm. ex Strid
*Dianthus petraeus* Waldst. & Kit.
*D.* sect. “*Plumaria*” Opiz
*Dianthus superbus* L.

**Table 2 plants-12-02094-t002:** The chemical composition of the diethyl-ether washings of fresh aerial parts and/or flowers of *D. carthusianorum* (**1a**–**1d**), *D. deltoides* (**2a**–**2c**), *D. giganteus* subsp. *banaticus* (**3**), *D. integer* subsp. *minutiflorus* (**4**), *D. petraeus* (**5a**–**5c**), *D. superbus* (**6a**–**6b**), and *P. prolifera* (**7**) from Serbia.

RI ^[a]^	Compound	Class ^[b]^	Samples ^[c]^															ID ^[d]^
			1a	1b	1c	1d	2a	2b	2c	3	4	5a	5b	5c	6a	6b	7	
802	Hexanal	G													tr			MS, RI, CoI
852	(*Z*)-Hex-3-en-1-ol	G													tr	tr		MS, RI, CoI
900	Nonane	A													tr			MS, RI, CoI
960	(*Z*)-Hept-2-enal	G													tr			MS, RI, CoI
1000	Decane	A													tr			MS, RI, CoI
1001	Octanal	G														tr		MS, RI, CoI
1005	(*Z*)-Hex-3-en-1-yl acetate	G														tr		MS, RI
1036	Benzyl alcohol	O													0.1	tr		MS, RI, CoI, Si
1047	Phenylacetaldehyde	O						tr	tr									MS, RI, CoI
1049	(*E*)-β-Ocimene	M									0.5							MS, RI
1063	Unidentified constituent ^[e]^								5.3							2.8		
1069	Octan-1-ol	G													tr			MS, RI, Si
1100	Undecane	A													tr			MS, RI, CoI
1106	Nonanal	G													tr	tr		MS, RI, CoI
1120	Maltol	O													0.4	18.0		MS, RI, Si
1122	2-Phenylethan-1-ol	ALC						tr										MS, RI, CoI, Si
1164	*neoiso*-Isopulegol	M					tr											MS, RI, CoI, Si
1190	Dodec-1-ene	AE													tr			MS, RI
1200	Dodecane	A								tr					tr	tr		MS, RI, CoI
1206	Decanal	G													0.2	tr		MS, RI, CoI
1274	3-Methyldodecane	A													tr			MS, RI
1300	Tridecane	A								tr					tr			MS, RI, CoI
1309	Undecanal	G													tr			MS, RI
1340	Methyl *o*-methoxybenzoate	E													tr			MS, RI
1367	(*E*)-Undec-2-enal	AL													tr			MS, RI
1374	Butyl benzoate	BZ					tr											MS, RI
1379	α-Copaene	S									tr							MS, RI
1400	Tetradecane	A													tr			MS, RI, CoI
1408	(*Z*)-Caryophyllene	S													0.1			MS, RI
1425	(*E*)-Caryophyllene	S									2.9	1.8		0.3	6.5	tr		MS, RI, CoI
1457	α-Humulene	S									tr				0.3			MS, RI
1465	*cis*-Muurola-4(14),5-diene	S									tr							MS, RI
1484	Germacrene D	S									tr				0.1			MS, RI, CoI
1494	1-(4-Hydroxy-3-methoxphenyl)ethan-1-one (syn. apocynin)	O													0.1			MS, RI
1500	Pentadecane	A													tr			MS, RI, CoI
1501	Bicyclogermacrene	S									tr							MS, RI, CoI
1502	(*E*)-Methyl isoeugenol	O													0.1			MS, RI, Si
1513	Tridecanal	AL								tr					tr			MS, RI
1527	δ-Cadinene	S									tr				tr			MS, RI
1568	(*E*)-Nerolidol	S									0.5							MS, RI, Si
1570	Vanillic acid	O					tr											MS, RI, Si
1575	Dodecanoic acid	AC													tr			MS, RI, Si
1581	Isovanillic acid	O													tr			MS, Si
1593	Caryophyllene oxide	S										tr			0.3			MS, RI, CoI
1600	Hexadecane	A								tr	tr				tr		tr	MS, RI, CoI
1616	Tetradecanal	AL								tr					tr			MS, RI
1681	Tetradecan-1-ol	ALC													tr			MS, RI, Si
1690	(*E*)-2,3-Dihydrofarnesol	S													0.4			MS, RI, Si
1699	Pentadecan-2-one	K								0.2				tr	tr			MS, RI, CoI
1700	Heptadecane	A								tr	tr				0.1		tr	MS, RI, CoI
1717	Pentadecanal	AL								tr	tr			tr	tr			MS, RI, CoI
1736	3-(4-Hydroxy-3-methoxyphenyl)propanoic acid (*syn*. dihydroferulic acid)	O					2.3											MS, RI, Si
1769	Benzyl benzoate	E					tr				4.2							MS, RI
1790	Octadec-1-ene	AE													tr			MS, RI
1800	Octadecane	A												tr	0.1		0.4	MS, RI, CoI
1820	Hexadecanal	AL								tr				tr	0.1			MS, RI, CoI
1839	Neophytadiene (isomer II)	D							tr									MS, RI
1846	Hexahydrofarnesyl acetone	O							tr									MS, RI
1867	Phenylethyl benzoate	BZ					tr				tr							MS, RI
1885	Hexadecan-1-ol	ALC												0.6	1.4			MS, RI, CoI, Si
1890	Nonadec-1-ene	AE										tr						MS, RI
1900	Nonadecane	A					tr	tr		tr	tr	tr			tr		tr	MS, RI, CoI
1903	Heptadecan-2-one	K								0.4				0.4	tr	tr		MS, RI, CoI
1922	Heptadecanal	AL							tr	0.1				tr	0.1			MS, RI, CoI
1970	Hexadecanoic acid	AC					0.3		tr	tr					0.2		tr	MS, RI, CoI, Si
1987	Heptadecan-1-ol	AE													tr			MS, RI, Si
2000	Eicosane	A	tr				tr	tr		0.1	tr	tr		0.1	0.1		0.4	MS, RI, CoI
2006	Hexadecyl acetate	E										tr		tr	0.1			MS, RI
2007	Octadecan-2-one	K															tr	MS, RI
2024	Octadecanal	AL					tr			tr	tr			tr	tr			MS, RI, CoI
2027	(*E*,*E*)-Geranyl linalool	D										0.3		0.4	0.6			MS, RI
2057	(*Z*,*Z*)-9,12-Octadecadien-1-ol	ALC												0.3	1.6			MS, RI, Si
2063	2-Methyleicosane	A					0.1	tr		tr	tr				0.2	tr	tr	MS, RI
2074	(*E*)-Heneicos-10-ene	AE								tr	0.1	0.5		0.2	0.3			MS, DMDS
2089	Octadecan-1-ol	ALC													0.1			MS, RI, Si
2100	Heneicosane	A	tr		tr	tr	0.3	tr	tr	0.2	0.5	2.1		0.8	0.5	tr	tr	MS, RI, CoI
2103	Methyl linoleate	E													0.1			MS, RI
2105	γ-Hexadecalactone	E					tr				0.4	3.3		2.4	tr			MS, RI
2113	(*E*)-Phytol	D							1.9						0.1			MS, RI, CoI, Si
2127	Nonadecanal	AL	tr			tr	tr	tr		0.1	tr	tr		0.3	0.2	tr		MS, RI
2143	(*Z*,*Z*)-9,12-Octadecadienoic acid	AC													0.1			MS, RI, Si
2149	(*Z*,*Z*,*Z*)-9,12,15-Octadecatrienoic acid	AC													0.3			MS, RI, Si
2163	2-Methylheneicosane	A										tr			tr			MS, RI
2173	(*E*)-Docos-10-ene	AE					0.1				tr			tr	0.2			MS, DMDS
2173	3-Methylheneicosane	A													tr		tr	MS, RI
2191	Ethyl octadecanoate	E					tr											MS, RI
2192	Nonadecan-1-ol	AE				tr									tr		tr	MS, RI, Si
2190	Hexadecanamide	FAD								tr								MS, RI
2200	Docosane	A	tr		tr	tr	tr	tr	tr	0.2	tr	tr		0.1	0.1	tr	0.3	MS, RI, CoI
2210	Eicosan-2-one	K															tr	MS, RI
2213	Dodecyl benzoate	BZ				tr											tr	MS, RI, CoI
2229	Eicosanal	AL	tr		tr	tr	tr	tr	tr	0.1	tr			tr	tr	tr	tr	MS, RI
2235	(*Z*)-Tricos-10-ene	AE													0.1			MS, DMDS
2264	2-Methyldocosane	A	tr		tr		0.2	tr		tr	0.3	tr		tr	0.2	tr	tr	MS, RI
2275	(*E*)-Tricos-10-ene ^[f]^	AE					tr	tr			1.4	1.0		0.7	3.1	tr		MS, RI, DMDS
2275	(*E*)-Tricos-9-ene ^[f]^	AE																MS, RI, DMDS
2292	Tricos-1-ene	AE								0.1					tr		tr	MS, RI
2296	Eicosan-1-ol	ALC													tr	tr		MS, RI, Si
2300	Tricosane	A	3.7	tr	1.8	0.2	0.4	0.3	tr	1.2	2.4	2.9	tr	1.5	3.4	0.2	0.2	MS, RI, CoI
2312	Heneicosan-2-one	K				tr				tr								MS, RI
2315	Tridecyl benzoate	BZ				tr									tr		tr	MS, RI, CoI
2332	Heneicosanal	AL	tr		tr	tr	tr	tr	tr	0.2	0.3	0.4		0.3	0.5	tr		MS, RI
2365	2-Methyltricosane	A	tr		tr					0.1		tr			tr			MS, RI
2373	3-Methyltricosane	A					0.2	tr			tr			0.1	tr		tr	MS, RI
2382	Hexadecyl hexanoate	E													tr			MS
2390	Tetracos-1-ene	AE				0.1			tr	tr	0.7	tr		tr	0.3	tr	0.1	MS, RI
2391	Ethyl eicosanoate	E					0.1											MS, RI
2395	Octadecanamide	FAD								tr								MS, RI
2400	Tetracosane	A	0.4	tr	tr	0.2	tr			tr	0.2	tr	tr	0.1	0.3	0.1	0.3	MS, RI, CoI
2414	Docosan-2-one	K															tr	MS, RI
2418	Tetradecyl benzoate	BZ				tr	0.1	tr		tr				tr	0.1		tr	MS, RI, CoI
2434	Docosanal	AL	0.5		tr	tr	0.1	0.3	0.4	0.1		tr		0.1	0.1	0.1	0.1	MS, RI
2435	(*Z*)-Pentacos-10-ene ^[f]^	AE								0.1	0.7			tr	0.6			MS, DMDS
2435	(*Z*)-Pentacos-9-ene ^[f]^	AE																MS, DMDS
2436	Octadecyl 3-methylbutanoate	E					0.1											MS, RI, CoI
2440	(2*E*,6*E*)-2,6-Farnesyl benzoate	BZ													0.2			MS, RI, CoI
2464	2-Methyltetracosane	A	tr		tr		0.1	tr			tr	tr		0.1	0.2		tr	MS, RI
2474	3-Methyltetracosane	A								0.7								MS, RI
2475	(*E*)-Pentacos-10-ene ^[f]^	AE			tr		tr				7.3	1.1		0.9	2.8			MS, DMDS
2475	(*E*)-Pentacos-9-ene ^[f]^	AE																MS, DMDS
2492	Pentacos-1-ene	AE								0.1								MS, RI
2493	12-Methyltetradecyl benzoate	BZ	tr				0.1	tr										**NEW**
2499	Docosan-1-ol	ALC													0.7			MS, RI, Si
2500	Pentacosane	A	5.9	0.5	5.2	2.5	1.0	0.5	1.2	3.6	8.2	4.4	1.6	2.7	4.0	1.0	1.1	MS, RI, CoI
2510	(*E*)-Pentacos-2-ene	AE								tr								MS, RI
2516	Tricosan-2-one	K															tr	MS, RI
2521	Pentadecyl benzoate	BZ				tr	0.1	tr		tr				tr			tr	MS, RI, CoI
2525	Methyl docosanoate	E								tr								MS, RI, CoI
2536	Tricosanal	AL	tr	tr	tr	tr	tr	tr	tr	0.1	tr	tr		0.1	0.1	tr	0.1	MS, RI
2563	2-Methylpentacosane	A	tr		tr					0.1	tr	0.2		tr	tr		tr	MS, RI
2574	3-Methylpentacosane	A	tr				0.1	tr		0.3	1.4			0.1	0.3	tr	0.1	MS, RI
2593	Hexacos-1-ene	AE	tr			tr	tr	tr	tr	0.1	0.3	tr	tr	0.1	0.3	0.1	tr	MS, RI
2596	13-Methylpentadecyl benzoate	BZ					tr	tr										**NEW**
2600	Hexacosane	A	1.1	tr	1.4	0.3	tr	tr	tr	1.1	0.4	0.3	tr	0.3	0.3	0.2	0.4	MS, RI, CoI
2610	(*E*)-Hexacos-2-ene	AE												tr	tr		tr	MS, RI
2612	1-Docosyl acetate	E					tr	tr				tr			0.1			MS, RI
2617	Tetracosan-2-one	K								tr								MS, RI
2625	Hexadecyl benzoate	BZ					tr	tr		tr	tr	tr		0.2	0.1			MS, RI, CoI
2637	(*Z*)-Heptacos-10-ene ^[f]^	AE						0.4			0.7	tr			0.5	0.3		MS, DMDS
2637	(*Z*)-Heptacos-9-ene ^[f]^	AE																MS, DMDS
2638	Tetracosanal	AL	1.0	0.8	tr	0.1	0.2		1.4	0.4		tr		0.3			0.3	MS, RI
2643	Eicosyl 3-methylbutanoate	E					0.4	0.4										MS, RI
2644	3-Methylbutyl eicosanoate	E					tr											MS, RI
2663	2-Methylhexacosane	A	0.9		1.0	tr	0.5	0.8	tr	3.9	1.8	0.4	tr	0.4	0.5		0.6	MS, RI
2674	3-Methylhexacosane	A				tr				0.8							tr	MS, RI
2675	(*E*)-Heptacos-10-ene ^[f]^	AE	tr		tr		tr	tr			2.0	0.7		1.2	1.5	tr	tr	MS, DMDS
2675	(*E*)-Heptacos-9-ene ^[f]^	AE																MS, DMDS
2675	(*E*)-Heptacos-8-ene ^[f]^	AE																MS, DMDS
2699	14-Methylhexadecyl benzoate	BZ					tr	tr										**NEW**
2700	Heptacosane	A	21.4	3.9	42.3	2.3	1.6	1.7	3.1	16.4	8.4	6.8	4.2	6.7	4.9	3.2	4.5	MS, RI, CoI
2718	Pentacosan-2-one	K		tr		0.3			tr	0.2		tr	tr	0.4		0.2	tr	MS, RI
2728	Heptadecyl benzoate	BZ					tr							tr				MS, RI, CoI
2730	Methyl tetracosanoate	E													tr			MS, RI
2739	Pentacosanal	AL	tr	tr	tr	0.1	tr	tr	tr			tr		0.2	tr	0.1	0.2	MS, RI
2740	Eicosyl tiglate	E					tr	tr					tr					**NEW**
2764	2-Methylheptacosane	A	tr		tr		tr	tr		0.2	tr	tr		0.1	tr	tr	tr	MS, RI
2775	3-Methylheptacosane	A	0.6			tr	2.1	2.7	tr	0.9	1.8	0.4		0.4	0.7	tr	1.6	MS, RI
2790	Octacos-1-ene	AE				tr		0.2	tr			tr	tr	tr	0.2	0.1	tr	MS, RI
2800	Octacosane	A	0.9	tr	1.9	0.5	0.4	0.3	tr	1.3	0.2	0.4	tr	0.6	0.3	0.3	1.4	MS, RI, CoI
2813	Tetracosyl acetate	E					tr	tr				tr			0.1		tr	MS, RI
2821	Hexacosan-2-one	K															tr	MS, RI
2832	Octadecyl benzoate	BZ					0.6	0.4						tr	tr			MS, RI, CoI
2839	(*Z*)-Nonacos-10-ene	AE									0.5	tr			0.5			MS, DMDS
2840	*all*-(*E*)-Squalene	T				0.5	tr	tr	0.6					tr		0.2	0.6	MS, RI, CoI
2843	Hexacosanal	AL	1.4	2.7	0.5	3.0	3.2	3.5	9.4	tr		0.3	0.8	1.0		2.3	5.5	MS, RI
2864	2-Methyloctacosane	A	1.2	tr	1.4	tr	2.5	2.5	tr	3.6	1.7	0.4		0.6	0.7	tr	1.8	MS, RI
2874	3-Methyloctacosane	A															0.3	MS, RI
2875	(*E*)-Nonacos-10-ene ^[f]^	AE	tr		tr		1.2	1.8	10.0	1.1	3.6	1.2		2.2	2.5	tr	0.6	MS, DMDS
2875	(*E*)-Nonacos-9-ene ^[f]^	AE																MS, DMDS
2890	2-Phenylethyl octadecanoate	E					tr											MS, RI
2900	Nonacosane	A	9.1	14.2	12.7	7.6	13.9	15.3	26.4	8.2	3.5	4.3	8.6	6.7	5.6	11.7	10.4	MS, RI, CoI
2909	Hexacosan-1-ol	ALC		0.6		15.7									tr		10.9	MS, RI, Si
2925	Heptacosan-2-one	K		tr				tr	1.1			0.5	tr	1.1	0.3	0.6	0.7	MS, RI
2934	Methyl hexacosanoate	E		tr		tr									0.3			MS, RI
2936	Nonadecyl benzoate	BZ					tr	tr			tr	tr						MS, RI, CoI
2945	Heptacosanal	AL	tr	tr	tr	0.5	tr	0.4		0.2		tr		0.3	0.1	0.4	0.4	MS, RI
2956	Tetradecyl tetradecanoate	E										0.5	tr	0.6	tr			MS, RI
2964	2-Methylnonacosane	A	1.4		1.6		tr	tr		0.3	tr		tr		tr	tr	tr	MS, RI
2969	Hexadecyl dodecanoate	E												0.1	tr			MS, RI
2974	3-Methylnonacosane	A					8.9	9.4	0.8	1.7	1.9	0.5		0.6	1.3	tr	2.6	MS, RI
2975	(*E*)-Triacont-10-ene	AE	tr			1.3	0.6	0.8	tr		0.2			0.2	0.4	tr	0.7	MS, DMDS
2984	Hexyl docosanonate	E													tr			MS, RI, CoI
2992	Triacont-1-ene	AE				tr			tr									MS, RI
3000	Triacontane	A	0.6	0.9	0.8	2.4	1.1	0.9		0.6	0.2	0.2	tr	0.2	0.6	1.2	1.2	MS, RI, CoI
3016	Hexacosyl acetate	E	tr	tr	tr	0.3		tr	0.6	0.1	tr	tr		0.1	0.1		0.2	MS, RI
3017	Tetracosanamide	FAD								tr								MS
3028	Octacosan-2-one	K				tr				tr				tr				MS, RI
3040	Eicosyl benzoate	BZ			0.3		4.1	8.1	tr	tr	0.7	0.1	tr	tr				MS, RI, CoI
3042	(*Z*)-Hentriacont-10-ene	AE	tr				2.0	0.3		0.2	1.0	tr			0.3			MS, DMDS
3049	Octacosanal	AL	2.4	2.6	1.9	1.1	2.0	1.5	1.7	0.7		0.9	1.8	1.9	0.6	6.5	0.7	MS, RI
3063	2-Methyltriacontane	A	2.8	tr	3.2	tr	4.9	5.6	2.1		2.3	0.3	tr	0.5	1.0	0.1	0.7	MS, RI
3073	3-Methyltriacontane	A								5.2							tr	MS, RI
3078	(*E*)-Hentriacont-10-ene ^[f]^	AE	1.3		2.3		4.9	5.7	tr	2.8	9.5	4.1	tr	5.2	4.1	0.1		MS, DMDS
3078	(*E*)-Hentriacont-9-ene ^[f]^	AE																MS, DMDS
3100	Hentriacontane	A	9.9	5.9	13.5	44.4	18.4	13.5	6.3	6.0	4.8	4.1	5.0	3.2	11.7	17.1	8.3	MS, RI, CoI
3114	Octacosan-1-ol	E		tr		2.4	tr		5.5	tr				1.3		9.6	0.8	MS, RI, Si
3117	18-Methyleicosyl benzoate	BZ	tr				tr	tr			tr							**NEW**
3130	Nonacosan-2-one	K	tr	tr			tr	1.1	tr	0.4	tr	0.9	tr	1.6		tr	1.1	MS, RI
3140	Methyl octacosanoate	E			tr										tr			MS, RI
3142	α-Tocopherol	O				1.1									tr	1.0	tr	MS, RI, CoI, Si
3143	Heneicosyl benzoate	BZ					tr	1.1			tr			0.3	0.3			MS, RI, CoI
3144	(*Z*)-Dotriacont-10-ene	AE	tr			0.7	0.8	0.7	tr		0.2			tr	tr	0.6		MS, DMDS
3150	Dodecyl octadecanoate	E		tr													0.4	MS, RI
3151	Tetradecyl hexadecanoate	E										1.6		2.5	tr	tr		MS, RI
3152	Nonacosanal	AL	tr	tr	tr	tr											0.2	MS, RI
3152	Hexadecyl tetradecanoate	E										tr		0.3	0.2			MS, RI
3163	2-Methylhentriacontane	A								1.4								MS, RI
3173	3-Methylhentriacontane	A	1.7	tr	1.6		3.5	3.9	0.7		1.3	0.4		0.3	1.1	tr	0.6	MS, RI
3177	(*E*)-Dotriacont-10-ene	AE												0.3				MS, DMDS
3197	Nonacosane-12,14-dione	DK	tr	tr						0.3		1.4	6.6	tr	0.9	tr		MS, HZ, Si
3200	Dotriacontane	A	0.5	tr	0.4	1.6	0.4	0.2	tr	0.6	tr	0.4		0.4	0.4	0.5	0.5	MS, RI, CoI
3210	20-Methylheneicosyl benzoate	BZ					0.2	0.3			tr	tr		tr	0.1			MS, RI
3216	Octacosyl acetate	E	tr		0.9	0.1				0.4		tr		0.4	0.2	0.2	0.1	MS, RI
3232	Triacontan-2-one	K								tr								MS, RI
3233	Docos-15-en-1-yl benzoate	BZ	tr				0.3	0.3		tr	0.1	tr		tr	0.2			MS, RI
3247	Docosyl benzoate	BZ	0.5	tr	1.1	tr	3.3	6.5	1.5	0.5	2.6	0.6	tr	1.1	0.5		0.5	MS, RI, CoI
3254	Triacontanal	AL	0.4	1.1	0.6	0.3						tr	tr			0.9	0.2	MS, RI
3264	2-Methyldotriacontane	A	0.5				0.4	0.6	tr	1.5	0.2			0.1	0.1		0.6	MS, RI
3278	(*E*)-Tritriacont-10-ene ^[f]^	AE	tr		tr		0.6	tr	tr		tr	tr		0.2	0.1			MS, DMDS
3278	(*E*)-Tritriacont-9-ene ^[f]^	AE																MS, DMDS
3290	Tritriacont-1-ene	AE	tr					0.5							tr			MS, RI
3295	16-Hentriacontanone	K														5.7	tr	MS, RI
3297	Triacontane-14,16-dione	DK											tr		tr			**NEW**
3300	Tritriacontane	A	2.3	tr	2.2	2.7	0.9	0.6	0.3	1.2	0.2	0.4	1.1	0.3	0.8	0.5	0.2	MS, RI, CoI
3316	Triacontan-1-ol	AE					tr	tr	0.4			tr			tr	0.8		MS, RI, Si
3324	20-Methyldocosyl benzoate	BZ	tr		tr					tr		tr			tr			**NEW**
3334	Hentriacontan-2-one	K	tr	tr		0.3				0.5			tr	tr	tr		1.0	MS, RI
3337	5α-Stigmasta-7,22-dien-3β-ol	O					1.6	1.0	4.8						0.6	0.7	0.6	MS, RI
3350	Tricosyl benzoate	BZ	tr		tr			tr			tr				0.1			MS, RI, CoI
3350	Dodecyl eicosanoate	E															0.7	MS, RI
3351	Hexyl 24-methylpentacosanoate	E								tr	tr							MS, RI
3351	Tetradecyl octadecanoate	E		0.7		tr	tr			0.2		0.2		1.1	0.5		0.3	MS, RI
3352	Hexadecyl hexadecanoate	E		tr			0.2		1.7	0.2		0.8		1.7	0.7	0.7		MS, RI
3365	2-Methyltritriacontane	A	0.3		tr													MS, RI
3375	3-Methyltritriacontane	A					tr							0.4				MS, RI
3390	Hexyl hexacosanoate	E								tr	tr							MS, RI
3394	*N*-(2-phenylethyl)eicosanamide	FAD								0.7								MS
3396	Hentriacontane-14,16-dione	DK	22.7	53.0						11.8	6.9	41.0	64.5	31.0	13.5	5.7	1.1	MS, HZ, Si
3400	Tetratriacontane	A			tr													MS, RI, CoI
3401	Benzyl tetracosanoate	E	0.4	3.1						tr								MS, RI
3417	22-Methyltricosyl benzoate	BZ					tr	tr		0.6	tr				tr		tr	MS, RI
3427	21-Methyltricosyl benzoate	BZ						tr			tr			0.7	0.4			**NEW**
3453	Tetracosyl benzoate	BZ	tr	tr	tr	0.2	0.3	tr	tr		0.6	tr	tr		0.2	tr	0.8	MS, RI, CoI
3458	Dotriacontanal	AL	tr	0.6	tr	0.7			tr							0.4	0.5	MS, RI
3460	Hexyl 24-methylhexacosanoate	E							tr									MS, RI
3498	Dotriacontane-14,16-dione	DK											tr		tr			**NEW**
3500	Pentatriacontane	A	0.6	tr	0.4	tr	tr			0.3	tr	tr	tr		0.1		0.1	MS, RI, CoI
3508	Benzyl pentacosanoate	E							tr									MS, RI
3531	22-Methyltetracosyl benzoate	BZ					tr	tr			tr				tr		tr	**NEW**
3538	Tritriacontan-2-one	K		tr		0.2										0.1	1.1	MS, RI
3556	Pentacosyl benzoate	BZ				tr									tr		tr	MS, RI
3590	Hexyl octacosanoate	E			tr													MS, RI
3596	*N*-(2-phenylethyl)docosanamide	FAD								0.2								MS
3598	Tritriacontane-16,18-dione	DK	0.4	5.0						1.7	tr	6.7	4.8	0.6	2.8	4.6	18.6	MS, HZ, Si
3600	Hexatriacontane	A	tr															MS, RI, CoI
3610	Benzyl hexacosanoate	E	tr	1.5		tr				0.4			tr		tr			MS, RI
3625	24-Methylpentacosyl benzoate	BZ					tr				tr	tr		tr				MS, RI
3635	23-Methylpentacosyl benzoate	BZ									tr							**NEW**
3659	Benzyl 25-methylhexacosanoate	E		tr	tr	0.4			tr						tr			MS, RI
3660	Hexacosyl benzoate	BZ	tr				0.2	tr	tr		0.8		tr		0.1		0.3	MS, RI
3698	Tetratriacontane-16,18-dione														tr	tr		**NEW**
3700	Heptatriacontane	A	tr							tr								MS, RI, CoI
3738	Pentatriacontan-2-one	K															0.2	MS, RI
3739	24-Methylhexacosyl benzoate	BZ					tr				tr							**NEW**
3764	Hexadecyl eicosanoate	E			tr	tr						tr		0.4				MS, RI
3765	Heptacosyl benzoate	BZ				tr					0.2				0.1		tr	MS, RI
3798	Pentatriacontane-16,18-dione	DK													tr	tr	0.1	MS, PZ
3812	Benzyl octacosanoate	E	tr	0.7						0.1			tr				tr	MS, RI
3844	25-Methylheptacosyl benzoate	BZ									tr							**NEW**
3870	Octacosyl benzoate	BZ	tr	tr	tr	tr					tr	tr	tr	tr	0.2			MS, RI
3948	26-Methyloctacosyl benzoate	BZ									tr						tr	**NEW**
3950	Docosyl hexadecanoate	E			tr	0.2		tr		tr		tr			0.2		tr	MS, RI
3951	Eicosyl octadecanoate	E	tr		tr		0.3								0.1			MS, RI
3974	Nonacosyl benzoate	BZ									tr						tr	**NEW**
3980	Hexadecyl docosanoate	E								tr				0.3				MS, RI
	**Total identified [%]**		**96.8**	**97.8**	**99.0**	**94.3**	**92.2**	**94.1**	**87.2**	**87.2**	**90.6**	**98.8**	**99.0**	**93.3**	**94.1**	**98.9**	**88.2**	
	Alkanes	A	65.8	25.4	91.4	64.7	61.9	58.8	40.9	61.7	41.7	28.9	20.5	27.4	39.6	36.1	38.6	
	Alkenes	AE	1.3	tr	2.3	2.1	10.2	10.4	10.4	4.6	28.2	8.6	tr	12.5	14.0	2.0	1.4	
	Aldehydes	AL	5.7	7.8	3.0	5.8	5.5	5.7	12.9	2.0	0.3	1.6	2.6	4.5	2.0	10.7	8.2	
	Alcohols	ALC		0.6		18.1			5.5	tr				0.9	5.2	9.6	11.7	
	Benzoates	BZ	0.5	tr	1.4	0.2	9.3	16.7	1.5	1.1	5.0	0.7	tr	2.3	2.6	tr	1.6	
	Diterpenoids	D							1.9			0.3		0.4	0.7			
	Diketones	DK	23.1	58.0				tr		13.8	6.9	49.1	75.9	31.6	17.2	10.3	19.7	
	Esters	E	0.4	6.0	0.9	3.6	1.1	0.4	2.3	1.4	4.6	6.4	tr	9.9	2.7	0.9	2.7	
	Fatty acids	AC					0.3		tr	tr					0.6		tr	
	‘Green leaf’	G													tr	tr		
	Ketones	K	tr	tr		0.8	tr	1.1	1.1	1.7	tr	1.4	tr	3.5	0.3	6.6	4.1	
	Monoterpenoids	M					tr				0.5							
	Other compound classes	O				1.1	3.9	1.0	4.8						1.2	19.7	0.6	
	Other fatty acid derivatives	FAD								0.9								
	Sesquiterpenoids	S									3.4	1.8		0.3	7.7	tr		
	Triterpenoids	T				0.5	tr	tr	0.6					tr		0.2	0.6	
	Unidentified								5.3							2.8		

^[a]^ Retention indices determined experimentally on the DB-5MS column relative to a series of C_8_–C_40_ *n*-alkanes. ^[b]^ The abbreviations of the compound classes are given at the end of the table. ^[c]^ Values are means of three individual analyses; tr, trace amounts (<0.05%). ^[d]^ ID = Compound identification: MS, mass spectra matching; RI, retention indices matching with the literature data; CoI, co-injection with a pure reference compound; DMDS, identification by derivatization with dimethyl disulfide; HZ, identification by derivatization with hydrazine; Si, identification by derivatization with trimethylsilyl chloride. ^[e]^ Unidentified constituent: MS (EI), *m*/*z*(%) 130(5), 112(6), 73(100), 71(8), 69(6), 58(51), 57(46), 56(4), 55(12), 45(8), 43(11), 42(5), 41(6). ^[f]^ Alkenes with different double bond locations represented one peak in the GC chromatogram and for that reason, it was not possible to determine their distinct relative amount in the washings.

**Table 3 plants-12-02094-t003:** Plant material and relevant data.

Plant Species	Plant Part	Sample	Location	Voucher Number	Year
*D. carthusianorum*	flowers	**1a**	Stara Planina	16618	2022
	aerial parts	**1b**			2022
	flowers	**1c**	Kopaonik	16619	2022
	aerial parts	**1d**			2022
*D. deltoides*	flowers	**2a**	Šara	13675	2018
	flowers	**2b**	Lake Vlasina	16389	2018
	aerial parts	**2c**			2018
*D. giganteus* subsp. *banaticus*	flowers	**3**	Deliblatska peščara	16620	2020
*D. integer* subsp. *minutiflorus*	flowers	**4**	Šara	13674	2018
*D. petraeus*	flowers	**5a**	Stara Planina	16621	2022
	aerial parts	**5b**			2022
	flowers	**5c**	Suva Planina	13981	2018
*D. superbus*	flowers	**6a**	Lake Vlasina	16390	2018
	aerial parts	**6b**			2018
*P. prolifera*	flowers	**7**	Stara Planina	16267	2022

## Data Availability

Not applicable.

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
