# Peer review of "Wax Composition of Serbian Dianthus spp. (Caryophyllaceae): Identification of New Metabolites and Chemotaxonomic Implications†"

_plants, 2023, doi:10.3390/plants12112094_

Round 1

Reviewer 1 Report

The manuscript is interesting and can be acceptable after some revisions including

1. The introduction is very short and should be re-written with more documented data of these species especially chemical and biological data

2. aim of this work should be re-written in clear style

3. Table 1 can be inserted in the supporting data not in the manuscript

4. The authors should described the meanings of the abbreviations of the class of the compounds under table 2 (such as FAD, O, .....)

5. "Similar MS fragmentation patterns 126 of 19 constituents (RI values 1374, 2213, 2315, 2418, 2521, 2625, 2728, 2832, 2936, 3040, 3143, 127 3247, 3350, 3453, 3556, 3660, 3765, 3870, and 3974) suggested that these constituents represent homologous benzoates of long-chain saturated n-alcohols.".   Where these RI in Table 2. The authors must insert the RI values of all identified compounds in Table 2.

6. The authors wrote RI values of many compounds in the text in the discussion, this is good. But the RI values of all compounds in Table 2 must be inserted

7. n-chain, n-alcohols, ......          "n" should be italic in all manuscript

8. I think "Figure S17. Dendrogram of PCA..." should be inserted in the manuscript

9. All the synthesis section in the experimental section.     I think this section should be supported be a chemical chart of the synthesis chemical equation

10. The conclusion should be summarized

The English should be carefully revised since the manuscript include some errors.

Author Response

Dear Editor,

we are submitting a manuscript that has been revised according to the suggestions and comments of the reviewers. All changes made in the manuscript are highlighted in green. Please find below a point-by-point reply to the reviewers’ comments. We thank the reviewers for their constructive suggestions and think that the manuscript has been considerably improved as a result. We hope that it will now be suitable for publication at Plants.

Fabio Boylan

Reviewer 1

  1. 1. The introduction is very short and should be re-written with more documented data of these species especially chemical and biological data.

Answer: Thank you for your suggestion, we gladly added more available data regarding the chemical analysis and biological activity of the Dianthus taxa.

  1. 2. Aim of this work should be re-written in clear style.

Answer: Part of the introduction with the aim of the work is revised.

  1. 3. Table 1 can be inserted in the supporting data not in the manuscript.

Answer: We kindly suggest that Table 1 should remain as a part of the manuscript. We believe that Table 1 contains the key data relating to the taxonomy of the species investigated in this study. Thus, the placement of these in the Supplementary Information file would greatly reduce their visibility and decrease the overall value of the main manuscript.

  1. 4. The authors should described the meanings of the abbreviations of the class of the compounds under table 2 (such as FAD, O, .....).

Answer: Explanations of the abbreviations of the compound classes are given at the end of Table 2 (e.g. FAD was the abbreviation for other fatty acid derivatives, etc.). Please see the last rows of Table 2. Thus, no additional explanation of the meaning is necessary.

  1. 5. "Similar MS fragmentation patterns of 19 constituents (RI values 1374, 2213, 2315, 2418, 2521, 2625, 2728, 2832, 2936, 3040, 3143, 127 3247, 3350, 3453, 3556, 3660, 3765, 3870, and 3974) suggested that these constituents represent homologous benzoates of long-chain saturated n-alcohols." Where these RI in Table 2. The authors must insert the RI values of all identified compounds in Table 2.

Answer: RI data for all detected wax constituents were given in the first column of Table 2. Also, the RI data were given in Tables S1 and S2. Thus, no values are missing.

  1. 6. The authors wrote RI values of many compounds in the text in the discussion, this is good. But the RI values of all compounds in Table 2 must be inserted.

Answer: The reply to the reviewer’s comment was the same as the comments in point 5. No values are missing.

  1. 7. n-chain, n-alcohols, "n" should be italic in all manuscript.

Answer: Revised as suggested.

  1. 8. I think "Figure S17. Dendrogram of PCA..." should be inserted in the manuscript

Answer: This figure was added to the manuscript as suggested and included into discussion where appropriate.

  1. 9. All the synthesis section in the experimental section. I think this section should be supported be a chemical chart of the synthesis chemical equation

Answer: Thank you for your valuable suggestion. A figure with the chemical reactions utilized was added to the Supplementary material (Figure S19).

  1. 10. The conclusion should be summarized.

Answer: Done as suggested. The conclusions section was significantly shortened, now focusing only on the major findings of the work.

  1. 11. The English should be carefully revised since the manuscript include some errors.

Answer: The language was corrected and rechecked by the professor of Pharmacognosy at the School of Pharmacy and Pharmaceutical Sciences, Trinity College Dublin, one of the corresponding authors. All changes to the text are marked by highlights.

Reviewer 2 Report

a large and labor-intensive effort to extract, separate, and identify a huge variety of wax compounds from Dianthus, which allowed isolation and identification of unidentified compounds.
Multivariate processing using unsupervised techniques is not adequate to discriminate environmental from taxonomic factors.
The use of multivariate supervised techniques could have given more guidance related to chemotaxonomic classification.
The research project should be better designed in terms of multivariate data processing. Otherwise, all identification work proves to be of little use for chemotaxonomic purposes.

Author Response

Dear Editor,

we are submitting a manuscript that has been revised according to the suggestions and comments of the reviewers. All changes made in the manuscript are highlighted in green. Please find below a point-by-point reply to the reviewers’ comments. We thank the reviewers for their constructive suggestions and think that the manuscript has been considerably improved as a result. We hope that it will now be suitable for publication at Plants.

Fabio Boylan

Reviewer 2

  1. 12. line 355: PCA and AHC showed that environmental factors produce sufficient background noise to prevent the expected taxonomic classification, so person correlation is related to background noise i.e., the main latent information. Specific information extraction might be made.

Answer: Thank you for your valuable suggestion. We completely agree with the reviewer’s comment and we added a new part of the discussion which relates to this shortcoming of the applied MVA analyses to warn the reader. The following was added:

“However, the low discrimination between the majority of the samples, as visible from the bi-plot (Figure 5) obtained from the PCA could be the result of environmental factors producing sufficient background noise to prevent the expected taxonomic classification. Therefore, one should be rather cautious in reaching any chemotaxonomic conclusions from such analyses. We tried to overcome this limitation by subjecting supervised data to all MVA, more specifically, the contents of the constituents with a relative amount ≥ 2%, 3%, 5%, 10%, 15%, 20%, and 25% in at least one of the compared samples, with the aim of achieving a better chemotaxonomic classification. The obtained results were either identical or very similar to the ones presented in Figure 5 (the corresponding biplots are not shown for that reason). It follows that either other classification variables need to be introduced or a significantly higher number of samples (e.g., taxa) needs to be treated to reach the desired statistical end result. When comparing the dendrograms obtained from a molecular biology study [1] with ours, sample sizes do not allow a meaningful interpretation and this is planned to be expanded in future studies.”

  1. 13. line 503: Multivariate data processing with unsupervised techniques shows that environmental factors produce considerable background noise. Therefore, data processing with supervised techniques, such as linear discriminant analysis, allows the extraction of location- and plant part-specific information in order to achieve better chemotaxonomic classification.

Answer: The reply to the reviewer’s comment was the same as the comments in point 12.

  1. 14. line 547: MVA approach does not allow characterizing phenotypic plasticity of waxes.

Answer: The reply to the reviewer’s comment was the same as the comments in point 12.

Reviewer 3 Report

The research manuscript by Mladenović and colleagues presents a comprehensive analysis of the chemical profile of the waxes from Dianthus spp., employing a chemotaxonomic approach. 

The introduction of the manuscript effectively establishes a strong foundation for the study, providing relevant background information. Furthermore, the experimental design is well crafted to address the proposed objectives. Notably, the utilization of strategies such as derivatization and esterification is commendable, as they enabled the identification of a remarkable 87-99% of compounds present in the samples.

The discussion pertaining to the chemical profile section is exemplary, providing a thorough and lucid explanation of compound identification. However, it would be beneficial to include the structures of the newly isolated compounds within the manuscript. This addition would enhance the visual comprehension of the findings.

In regard to the discussion of the chemotaxonomic section, there is room for improvement to achieve a more comprehensive analysis. Exploring the statistical analysis in greater detail would be advantageous. Additionally, it would be valuable to compare the chemotaxonomic data with existing genetic information available in the literature for these species. Incorporating such a comparison would significantly enhance the discussion section, offering a more robust demonstration of the methodology's advantages in aiding the taxonomic classification of a challenging genus.

Author Response

Dear Editor,

we are submitting a manuscript that has been revised according to the suggestions and comments of the reviewers. All changes made in the manuscript are highlighted in green. Please find below a point-by-point reply to the reviewers’ comments. We thank the reviewers for their constructive suggestions and think that the manuscript has been considerably improved as a result. We hope that it will now be suitable for publication at Plants.

Fabio Boylan

Reviewer 3

  1. However, it would be beneficial to include the structures of the newly isolated compounds within the manuscript. This addition would enhance the visual comprehension of the findings.

Answer: Thank you for your valuable suggestion. Two figures (Figure 2 and 3) with the structures of the newly identified compounds were added.

  1. In regard to the discussion of the chemotaxonomic section, there is room for improvement to achieve a more comprehensive analysis. Exploring the statistical analysis in greater detail would be advantageous. Additionally, it would be valuable to compare the chemotaxonomic data with existing genetic information available in the literature for these species. Incorporating such a comparison would significantly enhance the discussion section, offering a more robust demonstration of the methodology's advantages in aiding the taxonomic classification of a challenging genus.

Answer: Thank you for your valuable suggestion. We agree with the reviewer’s comment and we added a new part of the discussion which relates to this possibility but at the moment the two data sets used (this study and a previous molecular biology one from reference [1]) are significantly different to allow a direct and meaningful comparison. The following was added:

“However, the low discrimination between the majority of the samples, as visible from the bi-plot (Figure 5) obtained from the PCA could be the result of environmental factors producing sufficient background noise to prevent the expected taxonomic classification. Therefore, one should be rather cautious in reaching any chemotaxonomic conclusions from such analyses. We tried to overcome this limitation by subjecting supervised data to all MVA, more specifically, the contents of the constituents with a relative amount ≥ 2%, 3%, 5%, 10%, 15%, 20%, and 25% in at least one of the compared samples, with the aim of achieving a better chemotaxonomic classification. The obtained results were either identical or very similar to the ones presented in Figure 5 (the corresponding biplots are not shown for that reason). It follows that either other classification variables need to be introduced or a significantly higher number of samples (e.g., taxa) needs to be treated to reach the desired statistical end result. When comparing the dendrograms obtained from a molecular biology study [1] with ours, sample sizes do not allow a meaningful interpretation and this is planned to be expanded in future studies.”

Round 2

Reviewer 1 Report

The authors did the required revisions and the manuscript can be accepted in the present form 

minor language revisions is required